# Exploring the Impact of Knowledge about the Human Papillomavirus and Its Vaccine on Perceived Benefits and Barriers to Human Papillomavirus Vaccination among Adults in the Western Region of Saudi Arabia

**DOI:** 10.3390/healthcare12141451

**Published:** 2024-07-20

**Authors:** Fahad T. Alsulami

**Affiliations:** Clinical Pharmacy Department, College of Pharmacy, Taif University, Taif 21944, Saudi Arabia; f.alsulami@tu.edu.sa

**Keywords:** HPV, HPV vaccine, perceived benefits, perceived barriers, knowledge, Saudi Arabia

## Abstract

Objective: To evaluate knowledge about HPV and its vaccine, additionally, to examine the effect of knowledge about HPV and its vaccine on perceived benefits and barriers to HPV vaccination among individuals in the western region of Saudi Arabia. Methods: A cross-sectional design was employed in the western region of Saudi Arabia through a self-administered web-based survey. The survey assessed knowledge, perceived benefits, and perceived barriers to HPV vaccination. Results: A total of 1149 eligible participants completed the survey. Participants exhibited limited knowledge of HPV and its vaccine, with an average total score of 4.76 out of 15. Over 80% of participants were unaware that HPV may not show symptoms, cannot lead to HIV, and is not treatable with antibiotics. Over half were unaware that HPV can cause cervical cancer, oral cancer, or genital warts. Unmarried and graduate-educated participants demonstrated greater knowledge. Perceived benefits were positively associated with knowledge levels, while perceived barriers were negatively associated with knowledge levels after controlling for other variables. Conclusions: This study highlights the need for education and healthcare efforts to raise knowledge about HPV and its vaccine in the western region of Saudi Arabia. Also, this study indicated that enhanced knowledge boosts positive attitudes towards HPV vaccination, while reducing perceived barriers, thereby increasing vaccination rates.

## 1. Introduction

The human papillomavirus (HPV) comprises more than 100 subtypes and is recognized as the most common sexually transmitted infection worldwide, affecting over 11% of the global population, according to estimates [1]. HPV is categorized into two groups: low-risk and high-risk HPV strains. Low-risk HPV strains, exemplified by HPV 6 and 11, lead to the development of genital warts on the vulva and penis and may also manifest in the mouth, throat, or around the anus in both men and women. On the other hand, high-risk strains of HPV, such as HPV 16 and 18, play a role in the development of cancers affecting the cervix, vulva, vagina, penis, and anus [2,3]. These high-risk strains are implicated in approximately 5% of global cancer cases [4]. Almost all cases of cervical cancer and a significant proportion of anogenital cancers are linked to HPV infections [4]. Over 620,000 women and 70,000 men were diagnosed with HPV-related cancer globally in 2019, highlighting a significant disparity in diagnosis rates between genders [5]. HPV is responsible for 99% of cervical cancers, 90% of anal cancers, 75% of vaginal cancers, and 63% of penile cancers [4]. Additionally, while oropharyngeal cancer has traditionally been associated with tobacco and alcohol use, approximately 70% of oropharyngeal cancer cases are now attributed to HPV infection [4]. Despite the lack of comprehensive data regarding the extent of HPV infection and its link to cancers in Saudi Arabia, data show that Saudi Arabia experiences over 350 new instances of cervical cancer each year, with a crude mortality rate of 1.22 per 100,000 women annually [6].

Vaccination against HPV is the most effective means of safeguarding both men and women against the majority of cancers associated with HPV. Notable HPV vaccines, such as Gardasil^®^, Gardasil-9^®^, and Cervarix^®^, have proven to be highly effective in preventing infection with HPV strains that cause genital warts and cancers [7,8,9]. Initially, the HPV vaccine was aimed at girls between 11 and 12 years old, with the option for additional vaccination for females aged 13 to 26 years. In 2018, the United States FDA extended the coverage to include both males and females up to the age of 45 [10].

In 2010, the Saudi Food and Drug Administration granted authorization for HPV vaccines, such as the Cervarix^®^ vaccine and the Gardasil^®^ vaccine, specifically intended for females aged between 11 and 26 years [11]. In 2018, the Saudi National Immunization Schedule included HPV vaccines in the regular vaccination regimen for females, while they remain optional for males [12,13]. The HPV vaccines can be obtained for free from public hospitals and healthcare units, or for a charge from certain private hospitals and clinics in Saudi Arabia. They can also be accessed during regular appointments at pediatric clinics and family medicine. Male and female individuals can receive the HPV vaccine upon request in Saudi Arabia.

Numerous studies have been conducted in developed and developing countries to assess public knowledge, awareness, and acceptance of HPV vaccines since their introduction. Knowledge levels of HPV and the HPV vaccine were reported to be low to moderate [14,15,16]. Several factors, including age, sex, education level, and marital status, have been identified as directly associated with knowledge levels of HPV and the HPV vaccine [17,18]. Knowledge levels of HPV and the HPV vaccine were found to be positively associated with the perceived benefits, effectiveness, and safety of the HPV vaccine, which in turn correlate with a higher acceptance rate of the vaccine [19,20]. Nevertheless, other studies failed to identify a substantial correlation between awareness and knowledge levels of HPV and its vaccine and the perceived benefits of HPV vaccination [21,22,23]. On the other hand, knowledge levels of HPV and its vaccine were not reported to be significantly associated with perceived barriers to HPV vaccination [19,20,23].

In Saudi Arabia, several studies have assessed knowledge levels and awareness of HPV and its vaccines among different populations in various regions, reporting low knowledge levels of HPV and its vaccine [14,24,25,26,27]. Various factors, such as age, marital status, education level, and income level, have been found to be significantly associated with knowledge of HPV and its vaccines in Saudi Arabia [14,27]. However, limited studies have assessed the effect of knowledge levels about HPV and the HPV vaccine on perceived benefits and barriers to HPV vaccination in Saudi Arabia, which is crucial for understanding the decision-making process regarding the acceptance of vaccination. Almutairi et al. (2019) found a positive correlation between attitude toward the HPV vaccine and knowledge about HPV among medical students in Saudi Arabia [28].

Consequently, this study was designed to assess the improvement in knowledge levels of HPV and the HPV vaccine and to examine the effect of knowledge levels about HPV and its vaccine on perceived benefits and barriers to HPV vaccination among individuals in the western region of Saudi Arabia. This study significantly contributes to the body of knowledge on HPV and vaccination perceptions by addressing a critical gap in understanding the factors influencing the perception of HPV vaccination benefits and barriers among adults in the western region of Saudi Arabia. Furthermore, it provides insights beyond the local context, shedding light on the relationship between knowledge about HPV and its vaccine and the perceived benefits and barriers to vaccination. Such insights are valuable not only for Saudi Arabia but also for broader contexts. By elucidating these factors, this study aims to facilitate the development of targeted strategies by healthcare practitioners, policymakers, and public health interventions. These strategies aim to improve positive HPV vaccination attitudes and reduce barriers to vaccination, ultimately reducing the burden of HPV-related diseases and promoting public health in Saudi Arabia and potentially elsewhere.

## 2. Materials and Methods

### 2.1. Participants

A self-administered web-based survey employing a nonprobability sample of individuals in the western region of Saudi Arabia (Makkah region) was conducted through Google Forms (https://www.google.com/forms/about/, accessed on 25 November 2023) for a quantitative cross-sectional study. In December 2023, the survey link was disseminated to participants via WhatsApp and Telegram groups, and assistance from college students was sought for further distribution. The inclusion criteria for participants were being 18 years or older, residing in the western region of Saudi Arabia, and being capable of providing informed consent. To achieve the study’s objectives, a minimum of 385 participants were needed, which was calculated based on a population size of 8,325,304, a 95% confidence interval, and a 5% margin of error.

### 2.2. Measures

The research survey utilized several validated tools, including a sociodemographic questionnaire, an instrument measuring knowledge of HPV and the HPV vaccine, and two instruments assessing perceived benefits and barriers to HPV vaccination. These tools were developed based on the literature and were reviewed by two academic experts. Adjustments were made in accordance with the feedback provided by the reviewers. To ensure validity, the questionnaire underwent face and content validity checks through a pilot test involving ten participants; however, their responses were excluded from the final analysis. The survey items were crafted to be straightforward and were administered in both English and Arabic to prevent any potential misunderstandings. Additionally, the reliability of the instruments measuring knowledge, perceived benefits, and perceived barriers was evaluated using Cronbach’s alpha test.

#### 2.2.1. Knowledge of HPV and the HPV Vaccine

Individuals were provided with a set of items concerning HPV and the HPV vaccine, employing a 14-question scale drawn from the literature (e.g., “HPV infection always has visible signs or symptoms”, “Men cannot be infected with HPV”, and “HPV can be passed on during sexual intercourse”) [29,30,31]. Participants responded “True, False, or Don’t Know”. A correct response earned a score of one, while incorrect or “Don’t Know” responses received a score of zero. The cumulative knowledge score ranged from 0 to 14, with a higher score indicating greater knowledge about HPV and the HPV vaccine. The internal consistency of the knowledge scale, assessed through Cronbach’s alpha, was determined to be 0.82.

#### 2.2.2. Perceived Benefits and Barriers to HPV Vaccination

The perceived benefits of HPV vaccination were evaluated using four items (e.g., “The HPV vaccine is effective in preventing HPV”) adapted from the literature [32,33]. Responses from participants were gathered through a 5-point Likert scale, where 1 represented strongly disagree, and 5 represented strongly agree. The cumulative mean score ranged from 1 to 5, with a higher score indicating that participants perceived high benefits of HPV vaccination. The internal consistency of the perceived benefits construct demonstrated reliability, measured by Cronbach’s alpha, with a value of 0.90.

Perceived barriers to HPV vaccination were evaluated by utilizing three statements, such as “The HPV vaccine is unsafe,” adapted from the published literature [33,34]. Participants expressed their views on these statements by utilizing a 5-point Likert scale, which ranged from 1 (indicating strongly disagree) to 5 (indicating strongly agree). The cumulative mean score ranges from 1 to 5, with a higher score reflecting participants who perceived high barriers to HPV vaccination. The internal consistency of the perceived barrier construct demonstrated reliability, as measured by Cronbach’s alpha, yielding a value of 0.83.

### 2.3. Data Analysis

The analyses of all data were conducted utilizing IBM^®^ Statistical Package for the Social Sciences (SPSS), version 29.0. The data analysis employed a descriptive approach to elucidate and summarize the dataset. Additionally, independent-samples t-tests and one-way ANOVA were carried out to investigate the variations in the mean HPV knowledge scores across various sociodemographic factors. Furthermore, multiple linear regression analyses were conducted to identify parameters associated with perceived benefits and perceived barriers to HPV vaccination, including age, sex, marital status, employment status, education level, monthly income level, and knowledge of HPV and the HPV vaccine. Multicollinearity issues among the independent variables were assessed using the variance inflation factor (VIF). The analyses were carried out using two-tailed tests, and statistical significance was determined by a *p*-value below 0.05.

## 3. Results

### 3.1. Respondent Profile

A total of 1401 participants completed the survey. Only 1149 participants were included in this study because they met the inclusion criteria. The mean age was 34 years, and approximately 81% of the participants were female. Approximately 58% of participants were married, and 60% of them were unemployed. Additionally, more than half of the participants had a bachelor’s degree (64.3%). Approximately 57% of the participants had a monthly income of less than SAR 7000 (Table 1).

### 3.2. Knowledge Levels of HPV and the HPV Vaccine

Regarding the knowledge levels of HPV and the HPV vaccine, participants exhibited low levels of knowledge. The mean total knowledge score was 4.19 ± 3.36 out of 14. More than 80% of the participants were unaware that HPV infection does not always have visible signs or symptoms, cannot cause HIV, and cannot be cured by taking antibiotics. Additionally, more than half of the participants did not know that HPV infection can lead to cervical cancer, oral cancer, or genital warts. Furthermore, more than half of the participants were not aware that men can be infected with HPV, that HPV is transmitted through sexual intercourse, and that a person with no symptoms can transmit the HPV infection (Table 2).

Independent-samples t-tests and one-way ANOVA were conducted to examine the mean differences in knowledge levels across various sociodemographic characteristics. Participants who were not married exhibited greater knowledge levels than those who were married (*p*-value < 0.001). Furthermore, participants with a higher education degree demonstrated higher knowledge levels than did those with a high school education or less, a diploma, or a bachelor’s degree (*p*-value < 0.001). Participants with a monthly income between SAR 7000 and 10,999 had higher knowledge levels than did those with a monthly income of less than SAR 7000 or those with a monthly income of SAR 11,000 or more (*p*-value = 0.008) (Table 3).

### 3.3. Perceived Benefits and Barriers of HPV Vaccination

The study participants perceived moderate-to-high benefits of HPV vaccination; the overall mean score for perceived benefits was 3.36 out of 5. More than 40% of them agreed or strongly agreed that the HPV vaccine works well and is effective in preventing HPV, genital warts, and HPV-related cancers (Table 4).

Additionally, study participants perceived moderate barriers to HPV vaccination; the total average score was 2.86 out of 5. Over 20% of the participants agreed or strongly agreed that the HPV vaccine may lead to long-term health problems. Furthermore, approximately 19% and 17% of the participants agreed or strongly agreed that the HPV vaccine is being promoted for profit by pharmaceutical companies and that the HPV vaccine is unsafe, respectively (Table 4).

### 3.4. Predictors of Perceived Benefits and Barriers of HPV Vaccination

First, multiple linear regression was conducted to identify parameters (including age, sex, marital status, employment status, education levels, monthly income levels, and knowledge levels of HPV and the HPV vaccine) associated with perceived benefits of HPV vaccination. The model was significant (*p*-value < 0.001) and explained approximately 12% of the variance in perceived benefits of HPV vaccination (R^2^ = 0.12). Additionally, the model was examined for multicollinearity issues between independent variables using the variance inflation factor (VIF). The VIF values ranged between 1.06 and 2.21, indicating that there was no high correlation between independent variables. The model indicated that only the knowledge level of HPV and the HPV vaccine was significantly associated with perceived benefits of HPV vaccination; participants with high knowledge about HPV and the HPV vaccine tended to perceive more benefits of HPV vaccination (*p* < 0.001) (Table 5).

Then, multiple linear regression was conducted to identify parameters (including age, sex, marital status, employment status, education level, monthly income level, and knowledge levels of HPV and the HPV vaccine) associated with perceived barriers to HPV vaccination. The model was found to be statistically significant with a *p*-value less than 0.001 and accounted for about 5% of the variance in perceived barriers to HPV vaccination (R^2^ = 0.05). Additionally, the model was assessed for multicollinearity among the independent variables using the variance inflation factor (VIF). The VIF values, which ranged from 1.06 to 2.21, indicated that there was no substantial correlation between the independent variables. Only knowledge level of HPV and the HPV vaccine was significantly associated with perceived barriers to HPV vaccination; participants with high knowledge about HPV and the HPV vaccine tended to perceive fewer barriers to HPV vaccination (*p*-value < 0.001) (Table 6).

## 4. Discussion

Saudi Arabia has a population of 10.7 million females aged 15 and above who face the possibility of developing cervical cancer. According to current estimates, 358 women are diagnosed with cervical cancer each year, of which 179 die from the disease. Cervical cancer ranks as the eighth most prevalent cancer among women in Saudi Arabia [6]. Currently, there are no available data on the HPV burden in Saudi Arabia’s general population; however, in Western Asia, to which Saudi Arabia belongs, around 2.5 percent of women in the overall population are estimated to be infected with cervical HPV at any given time [35]. These alarming numbers support the necessity for immediate and effective preventive measures, while also properly educating the population about the severity of the disease.

The aim of this research was to evaluate individuals’ knowledge of HPV and its vaccine within the western region of Saudi Arabia. Furthermore, this research investigated the effect of knowledge levels of HPV and the HPV vaccine on perceived benefits and barriers to HPV vaccination within that population.

The findings of the study indicate that overall knowledge levels regarding HPV and the HPV vaccine were suboptimal among the participants. More than fifty percent of the respondents were unaware that contracting HPV could result in cervical cancer, oral cancer, and genital warts. Additionally, more than four-fifths of them were unaware that HPV infection may not exhibit visible signs or symptoms, does not lead to HIV, and cannot be treated by taking antibiotics. These findings are consistent with other studies conducted across several regions in Saudi Arabia among different population groups. Alhusayn et al. [11] found that over 70% of parents in Riyadh City, Saudi Arabia, were not aware that HPV infection can lead to cervical cancer. Furthermore, Altamimi [26] found that about 56% of female college students at Hail University did not know that HPV infection is a risk factor for cervical cancer. Additionally, another study conducted in the Jazan region of Saudi Arabia found that the average knowledge score about HPV and its vaccine was 1.99 out of 10 [27]. Another study conducted among male and female college students at five health-related colleges in Riyadh City, Saudi Arabia, found that the overall mean knowledge score about HPV and its vaccine was 8.8 out of 17 [36]. Furthermore, a study conducted among females in the Makkah region of Saudi Arabia found that about 39% of them knew that HPV is a sexually transmitted infection [37]. Similar to other studies that assessed knowledge levels about HPV and its vaccine across different Arab countries, Al Alawi et al. [38] found that adults in Oman had a low knowledge level about HPV and the HPV vaccines, with only 22.6% of them having heard of HPV infection. Another study conducted in the Kingdom of Bahrain found that only 13.5% of the participants were aware of HPV infection [39]. Saqer et al. [40] observed that 14.5% of parents knew that HPV infection can cause cervical cancer in Sharjah City, United Arab Emirates.

Comparing these findings with studies conducted in European countries, Sidiropoulou et al. [41] found that about 90% of participants were aware of HPV, and the mean HPV knowledge score was 53.26 out of 100 in Greece. Another study found that about 71% of women in the United Kingdom were aware of HPV infection, with a mean knowledge score of 3.7 out of 8 [42]. In Switzerland, Schwendener et al. [43] noted that about 70% of young participants were knowledgeable about the HPV vaccine.

This disparity in HPV awareness and knowledge between Arab and European countries highlights the need for targeted educational campaigns. Ensuring individuals have accurate information about HPV and understanding the importance of HPV vaccination in protecting against HPV-related diseases is essential, especially in Arab countries, including Saudi Arabia. Addressing this gap is crucial for the effective prevention and control of HPV-related diseases in these regions. Without adequate knowledge and awareness of HPV and its vaccine, individuals may fail to take preventive measures such as vaccination, increasing their risk of HPV-related health complications [44,45]. Additionally, these gaps in knowledge may hinder efforts to prevent HPV-related diseases through vaccination and early detection. However, healthcare providers play a crucial role in increasing awareness and knowledge and addressing any misconceptions or concerns about HPV and its vaccine that individuals may have [46].

In terms of marital status, participants who were not married exhibited higher levels of knowledge than those who were married (*p*-value < 0.001). This finding is surprising and contrasts with other studies, such as a recent study by Turki et al. [37], where married individuals had a higher level of knowledge about HPV (*p*-value = 0.029). These differences can be attributed to variations in sample size, with this study having a larger study population. Additionally, their questionnaire was addressed only to female participants in one city in the western region of Saudi Arabia, while this study included a broader area, incorporating results from the entire western region. Although this study did not find significant gender effects, the inclusion of both genders allows for a more comprehensive analysis and understanding of the population’s knowledge about HPV.

Additionally, this study found that participants with a graduate education degree tended to have higher knowledge levels of HPV and the HPV vaccine (*p*-value < 0.001). This is further supported by the results of multiple studies [47,48]. Such findings emphasize the importance of spreading awareness through educational materials aiming to increase knowledge about HPV and the importance of HPV vaccination among individuals with low education levels in Saudi Arabia. Furthermore, this study found that participants with a high school education degree or less had higher knowledge levels about HPV and its vaccine compared with those with diploma degrees, which is inconsistent with another study [37]. This finding might be explained by the fact that many participants classified under the “high school or less” category might actually be current college students who have not yet completed their degree. College students are often exposed to updated health information and educational campaigns in the college environment, which likely contributed to their higher knowledge levels. Therefore, further research is necessary to gain a clearer understanding of this dynamic.

Additionally, this study revealed that participants with a monthly income ranging from SAR 7000 to 10,999 showed a significantly higher level of knowledge regarding HPV and its vaccine compared to those with incomes below SAR 7000 or above SAR 10,999. This finding might be explained by the fact that it is possible that public health campaigns and educational programs have been particularly effective in reaching the middle-income group compared with lower- and higher-income groups. Addressing socioeconomic disparities in healthcare access and education can contribute not only to increased awareness of HPV but also to better overall health outcomes. Consequently, additional studies are required for a better understanding of this dynamic.

There was widespread variation in beliefs regarding the efficacy of the HPV vaccine. Over 40% of participants agreed that the vaccine works well and is effective in preventing HPV, genital warts, and HPV-related cancer. These findings are consistent with another study where about 46% of Saudis in the eastern region believed that the HPV vaccine is effective against HPV [49]. However, approximately 19% and 17% of the participants agreed or strongly agreed that the HPV vaccine is being promoted for profit by pharmaceutical companies and that it is unsafe, respectively. These findings are supported by a study in which concern about the safety of the HPV vaccine was the most common barrier to receiving the vaccine among female college students in Saudi Arabia [26].

This study found that the knowledge level about HPV and its vaccine was significantly associated with perceived benefits and barriers of the HPV vaccine after controlling for other variables. Individuals with a high knowledge level about HPV and its vaccine tend to perceive more benefits of HPV vaccination. This finding is supported by a study in which knowledge levels of HPV and its vaccine positively correlated with perceived benefits of the HPV vaccination and cervical cancer screening among female Saudis [50]. This finding underscores the importance of educational campaigns and healthcare initiatives aimed at increasing awareness and understanding of HPV and its vaccination among the population, as it can directly influence attitudes and behaviors towards preventive measures such as vaccination and screening.

In addition, this study found that individuals with a high level of knowledge about HPV and its vaccine tend to perceive fewer barriers to HPV vaccination. This finding is supported by another study that found perceived barriers were significantly and negatively associated with knowledge scores about HPV in Northern Cyprus [20]. This finding highlights a crucial intersection between perceived barriers and knowledge levels in shaping vaccination behavior. This suggests that misconceptions or lack of information about HPV and the HPV vaccine may contribute to the perception of barriers. Addressing these knowledge gaps through educational initiatives and targeted interventions could potentially alleviate some of the barriers that individuals perceive, thus promoting better HPV vaccination uptake.

## 5. Limitations and Strengths

Despite the key insights generated from the study, a few limitations should be noted. First, the study was based on self-reported data, which can be affected by recall bias. This occurs when participants inaccurately remember their past earnings or the correct information about HPV and its vaccine. Additionally, social perception bias may occur when participants respond more favorably or unfavorably to questions related to benefit and barrier perceptions of HPV vaccination due to perceived social desirability. Third, the study’s cross-sectional design limits the potential to assess causality or sequential associations among variables. Furthermore, the study sample was limited to the western region of Saudi Arabia, which may not be generalizable to the entire population. Lastly, this study examined the effect of knowledge levels about HPV and its vaccine on the perceived benefits and barriers to HPV vaccination without estimating the impact of this knowledge on HPV vaccination behavior. Future research should aim to overcome these limitations by using longitudinal designs with wider and more varied representative samples. Nonetheless, this study is unique in addressing the level and perception of HPV knowledge, especially in the entire western region of Saudi Arabia. Additionally, this research had the advantage of obtaining a good sample size with appropriate sample selection to infer results representative of the chosen population. Finally, the use of an organized questionnaire to collect data enabled standardized collection and analysis of the data.

## 6. Conclusions

This study explored the factors influencing perceptions of HPV vaccination and assessed knowledge levels regarding HPV and its vaccine among adults in the western region of Saudi Arabia. The findings highlight the importance of targeted educational programs, improved healthcare provider recommendations, and improved architecture in promoting HPV vaccination in Saudi Arabia. Addressing these challenges can efficiently lessen the burden of HPV-related diseases while also improving overall population health outcomes.

Efforts should be focused on constructing comprehensive educational campaigns for the public, schools, and community activists. These campaigns should aim to increase awareness of HPV, showcase the benefits of vaccination, combat popular misconceptions, and address cultural and religious concerns. While this study reveals an association between higher knowledge levels and more positive perceptions of HPV vaccination, further research is needed to explore the causal pathways between knowledge acquisition and perception formation.

## Figures and Tables

**Table 1 healthcare-12-01451-t001:** Sociodemographic characteristics of the study participants (*n* = 1149).

Sociodemographic Characteristic	*n* (%)	Mean ± SD
**Age**		34.35 ± 12.84
**Gender**		
Male	220 (19.1%)	
Female	929 (80.9%)	
**Marital Status**		
Married	671 (58.4%)	
Not married	478 (42.6%)	
**Employment Status**		
Employed	460 (40%)	
Not employed *	689 (60%)	
**Education**		
High school or less	265 (23.1%)	
Diploma degree	74 (6.4%)	
Bachelor’s degree	739 (64.3%)	
Higher education degree	71 (6.2%)	
**Monthly Income Level**		
Less than SAR 7000	657 (57.2%)	
SAR 7000 to SAR 10,999	135 (11.7%)	
SAR 11,000 to SAR 14,999	175 (15.2%)	
SAR 15,000 or more	182 (15.8%)	

SAR: Saudi riyal. * Includes students, retirees, and those who were looking for a job.

**Table 2 healthcare-12-01451-t002:** Mean total knowledge score and number of respondents who answered questions about HPV and the HPV vaccine correctly.

HPV and the HPV Vaccine Knowledge Item	*n* (%)	Mean ± SD
HPV is very rare. (F)	185 (16.1%)	
HPV always has visible signs or symptoms. (F)	147 (12.8%)	
There are many types of HPV. (T)	502 (43.7%)	
HPV can cause cervical cancer. (T)	515 (44.8%)	
HPV can cause HIV/AIDS. (F)	197 (17.1%)	
HPV can be passed on during sexual intercourse. (T)	504 (43.9%)	
HPV can cause genital warts. (T)	541 (47.1%)	
Men cannot get HPV. (F)	472 (41.1%)	
HPV can be cured with antibiotics. (F)	150 (13.1%)	
HPV can cause oropharyngeal cancer. (T)	235 (20.5%)	
A person with no symptoms cannot transmit the HPV infection. (F)	333 (29%)	
The HPV vaccine requires at least 2 doses. (T)	321 (27.9%)	
The HPV vaccines offer protection against all sexually transmitted infections. (F)	206 (17.9%)	
The HPV vaccines are most effective if given to people at a younger age. (T)	509 (44.3%)	
**Overall Mean Knowledge Score**		4.19 ± 3.36

**Table 3 healthcare-12-01451-t003:** Mean differences in knowledge level across different sociodemographic characteristics.

Sociodemographic Characteristic	Mean	SD	*p*-Value
**Gender**			0.262
Male	3.96	3.64	
Female	4.25	3.29	
**Marital Status**			**<0.001**
Married	3.88	3.34	
Not married	4.63	3.34	
**Employment Status**			0.064
Employed	3.97	3.49	
Not employed *	4.34	3.26	
**Education**			**<0.001 ****
High school or less	4.12	3.21	
Diploma degree	3.42	2.90	
Bachelor’s degree	4.15	3.32	
Higher education degree	5.70	4.23	
**Monthly Income Level**			**0.** **008 ****
Less than SAR 7000	4.32	3.22	
SAR 7000 to SAR 10,999	4.40	3.46	
SAR 11,000 to SAR 14,999	3.39	3.19	
SAR 15,000 or more	4.33	3.80	

* Includes students, retirees, and those who were looking for a job. ** One-way ANOVA. SAR: Saudi riyal. Numbers in bold represent statistically significant items.

**Table 4 healthcare-12-01451-t004:** Perceived benefits and barriers of the HPV vaccination constructs.

Item	Strongly Disagree and Disagree *n* (%)	Neutral *n* (%)	Strongly Agree and Agree *n* (%)
**Perceived Benefits**			
1. The HPV vaccine works well.	110 (9.6%)	561 (48.8%)	478 (41.6%)
2. The HPV vaccine is effective in preventing HPV.	110 (9.6%)	521 (45.3%)	518 (45.1%)
3. The HPV vaccine is effective in preventing genital warts.	131 (11.4%)	538 (46.8%)	480 (41.8%)
4. The HPV vaccine is effective in preventing HPV-related cancers.	122 (10.6%)	543 (47.3%)	484 (42.1%)
**Perceived Barriers**			
1. The HPV vaccine is unsafe.	366 (31.8%)	580 (50.5%)	203 (17.7%)
2. The HPV vaccine may lead to long-term health problems.	310 (27%)	608 (52.9%)	231 (20.1%)
3 The HPV vaccine is being pushed to make money for pharmaceutical companies.	348 (30.3%)	581 (50.6%)	220 (19.2%)

**Table 5 healthcare-12-01451-t005:** Multiple linear regression to identify the predictors associated with perceived benefits of HPV vaccination.

Parameter	Perceived Benefits of HPV Vaccination		
	Beta *	*p*-Value	VIF **
**Age**	0.034	0.406	2.215
**Gender**			
Female	Reference		
Male	−0.024	0.439	1.227
**Marital Status**			
Not married	Reference		
Married	−0.024	0.533	1.837
**Employment Status**			
Not employed ***	Reference		
Employed	0.006	0.876	1.884
**Education**			
High school or less	Reference		
Diploma degree	0.048	0.120	1.252
Bachelor’s degree	0.064	0.065	1.535
Higher education degree	0.016	0.636	1.459
**Monthly Income Level**			
Less than SAR 7000	Reference		
SAR 7000 to SAR 10,999	−0.014	0.676	1.437
SAR 11,000 to SAR 14,999	−0.068	0.066	1.774
SAR 15,000 or more	0.024	0.548	2.104
**HPV and the HPV Vaccine Knowledge**	0.331	**<0.001**	1.061

* Standardized beta coefficients. ** Variance inflation factor. *** Includes students, retirees, and those who were looking for a job. SAR: Saudi riyal. Numbers in bold represent statistically significant items.

**Table 6 healthcare-12-01451-t006:** Multiple linear regression to identify the predictors associated with perceived barriers to HPV vaccination.

Parameter	Perceived Benefits of HPV Vaccination		
	Beta *	*p*-Value	VIF **
**Age**	0. 084	0. 052	2. 215
**Gender**			
Female	Reference		
Male	0. 001	0. 968	1. 227
**Marital Status**			
Not married	Reference		
Married	−0. 044	0. 261	1.837
**Employment Status**			
Not employed ***	Reference		
Employed	0.007	0. 851	1.884
**Education**			
High school or less	Reference		
Diploma degree	0. 011	0. 728	1.252
Bachelor’s degree	0. 007	0. 849	1.535
Higher education degree	0. 027	0. 432	1. 459
**Monthly Income Level**			
Less than SAR 7000	Reference		
SAR 7000 to SAR 10,999	−0. 007	0. 836	1. 437
SAR 11,000 to SAR 14,999	−0. 017	0. 666	1.774
SAR 15,000 or more	−0. 048	0. 248	2.104
**HPV and the HPV Vaccine Knowledge**	−0. 215	** <0.001 **	1. 061

* Standardized beta coefficients. ** Variance inflation factor. *** Includes students, retirees, and those who were looking for a job. SAR: Saudi riyal. Numbers in bold represent statistically significant items.

## Data Availability

This article contains all of the study’s data.

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
