# Peer review of "Exploring the Impact of Knowledge about the Human Papillomavirus and Its Vaccine on Perceived Benefits and Barriers to Human Papillomavirus Vaccination among Adults in the Western Region of Saudi Arabia"

_healthcare, 2024, doi:10.3390/healthcare12141451_

Round 1

Reviewer 1 Report

Comments and Suggestions for Authors

Thank you for the opportunity to review the manuscript “Exploring the Impact of HPV and Its Vaccine Knowledge on Perceived Benefits and Barriers to HPV Vaccination Among Adults in the Western Region of Saudi Arabia” (ID healthcare-2999069).

Congratulations to the author! The manuscript is well designed, well written and the findings are presented and discussed appropriately. The final considerations are clear and precise. I have just one suggestion to the author: include a map clearly indicating to readers where Saudi Arabia is in the Middle East, Western Asia. I recommend the publication of this manuscript in the Healthcare.

Author Response

Please Look at the attached file

Reviewer 2 Report

Comments and Suggestions for Authors

The manuscript entitled "Exploring the Impact of HPV and Its Vaccine Knowledge on Perceived Benefits and Barriers to HPV Vaccination Among Adults in the Western Region of Saudi Arabia" by Fahad T. Alsulami is well written and presented.

The study design and approach to the data analysis was scientifically sound. The manuscript is also well referenced and is commendable for a single author study and publication.

I would have liked to see more consideration of legitimate concerns about the safety or adverse effects of the HPV vaccine and a critical assessment of the controversies surrounding vaccination in general. 

The data on whether HPV vaccination has resulted in a decrease in cervical cancer is limited and should be discussed more critically, see this reference and other related studies it cites in other countries - https://www.ncbi.nlm.nih.gov/pmc/articles/PMC9859059/#B25.

Also, HPV screening, early detection and treatment has been shown to be efficacious, so it would have been good to include a discussion on this within the manuscript.  What are the rates of HPV screening, early detection and treatment in Saudi Arabia and has this program been effective.  Could it be made more effective and what are the pros and cons of this older method of tackling the issue versus vaccination?  How do the costs compare? 

Currently the manuscript is very biased towards vaccination and assumes that negative attitudes to vaccination or vaccine hesitancy is illegitimate.  The side effects of vaccines are real and tend to be under-reported.  Informed consent involves providing accurate information about possible side effects, so this needs to be addressed and discussed more transparently within the manuscript and alternatives discussed more thoroughly - see https://www.ncbi.nlm.nih.gov/pmc/articles/PMC5967601/.  What is the risk benefit assessment of the vaccine in different populations and how accurate / unbiased is this data?  How does this compare to the older screening method?  What are the alternatives for those who may legitimately have a higher risk of adverse effects from the vaccine and how can public education programs tackle this and provide a more comprehensive and honest/transparent information so that there is true informed consent to the vaccine.  The author should make more of an effort to consider what could be legitimate concerns about the vaccine and provide a more balanced, unbiased analysis.

Author Response

Please Look at the attached file

Reviewer 3 Report

Comments and Suggestions for Authors

Please check the comments in the file attached.

Author Response

Please Look at the attached file

Reviewer 4 Report

Comments and Suggestions for Authors

The authors have done a commendable job in highlighting the importance of education in raising awareness about highly infectious diseases such as HPV. The study effectively demonstrates the impact of education on disease awareness and prevention among educated cohorts. The questions designed for participants are impressive. However, several points need to be addressed to enhance the manuscript's quality for the audience.

1.        There is considerable repetition throughout the manuscript that needs to be revised to improve clarity and conciseness.

2.        In the methods section (2.2.1 to 2.2.2), presenting the questions in a table format could help avoid confusion.

3.        In Table 2, the statement “The HPV vaccine requires at least 2 doses. (T)” is outdated given the availability of single-dose vaccines. Please update this information throughout the manuscript and add relevant references to avoid confusion, such as: The Lancet.

4.        The authors should explain why individuals with "High school or less" education scored better than those with a diploma degree and almost as well as those with a bachelor's degree.

5.        Similarly, an explanation is needed for why the "less monthly income" group achieved the best scores, as this is counterintuitive.

6.        Table 5 should be explained in layman's terms to make it accessible to a broader audience.

7.        The authors should compare their data with global data to determine if their findings are applicable worldwide or specific to the Saudi Arabian population.

Additionally, I would like to know how the authors plan to address myths such as the belief that the HPV vaccine may cause long-term health problems and is being promoted for profit by pharmaceutical companies.

Furthermore, it would be beneficial to include more critical thinking questions in the questionnaire and invite participants to a seminar or webinar post-questionnaire to better understand vaccines, dispel misconceptions, and correct any misinformation. Moreover, it would be beneficial to explore whether awareness programs would be effective regardless of participants' education status.

Overall, while the study is impressive, addressing these points will significantly enhance its impact and clarity for the readership.

Comments on the Quality of English Language

The English is generally good, with only minor revisions needed to improve clarity and readability.

Author Response

Please Look at the attached file

Round 2

Reviewer 3 Report

Comments and Suggestions for Authors

Thank you for providing a revised version of the manuscript. I think the paper has improved significantly and I only have a few minor comments. You may regard all other comments as resolved satisfactorily. I appreciate the effort you have put into addressing my questions and concerns.

In response to your numbering from the Response Letter:

8.       I understand your interpretation of the term “unemployed” departs from the OECD definition and the labour force statistics of Saudi Arabia. I wonder if this definition was clear to the respondents when filling in the survey. For completeness and transparency sake you may want to consider clarifying this aspect in a footnote in your article.

10.   In the underlying paragraph and subsection you make a point of characterising the knowledge level on HPV vaccination as low. In my review report 1 I raise the point of trying to put the knowledge level you record in your survey into perspective by either comparing it to knowledge levels about vaccinations against other viral infections, or alternatively to knowledge levels of HPV vaccinations in other geographical regions. You argue that these types of comparison fall outside the scope of the current study. I however believe that a comparison, drawn from other relevant literature, may be insightful in putting your results into perspective. This information could potentially increase the relevance of your results, if, for example, knowledge levels on HPV vaccination in Western Saudi Arabia are much lower than in other geographic areas in the Arab region / Asia / globally.

19.   This issue concerns the degree towards which your survey may be prone to recall bias and social perception bias. You acknowledge that your study – as many other survey studies – may be prone to these biases. If possible, I would invite you to be more specific about what respondents may recall incorrectly (would that be their earnings or their knowledge level on HPV vaccination?) and how exactly social desirability factors into your survey. Which question could be prone to social desirability bias and to what effect?

Editorial comments, ordered as I come across in the paper, not in importance.

2.       I welcome the addition to the underlying sentence you propose. In the revised manuscript I do not see the addition proposed in your review letter, though. Could you double check if the additional statement (“highlighting a significant disparity in diagnosis rates between genders"), which I think is a valuable qualification of the numbers presented, will be part of the actual manuascript?

Author Response

Thank you
